computational chemistry/biochemistry/
spectroscopy

DFT, vibrational spectroscopy, amyrins, SEM,
differential scanning calorimetry, Raman

**Authors for correspondence:**
Rafael C. González-Cano
e-mail: rafacano@uma.es
Antonio Heredia
e-mail: heredia@uma.es

This article has been edited by the Royal Society of Chemistry, including the commissioning, peer review process and editorial aspects up to the point of acceptance.

# Structural analysis of mixed α- and β-amyrin samples

Luz D. M. Gómez-Pulido[1], Rafael C. González-Cano[2], José J. Benítez[4], Eva Domínguez[1] and Antonio Heredia[3]

[1]IHSM La Mayora, Departamento de Mejora Genética y Biotecnología, Consejo Superior de Investigaciones Científicas, E-29750 Algarrobo-Costa, Málaga, Spain
[2]Departamento de Química Física, Facultad de Ciencias,
[3]IHSM La Mayora, Departamento de Biología Molecular y Bioquímica, Universidad de Málaga, E-29071 Málaga, Spain
[4]Instituto de Ciencia de Materiales de Sevilla. Centro Mixto CSIC-Universidad de Sevilla, E-41092, Sevilla, Spain

 LDMG-P, 0000-0003-4945-2377; RCG-C, 0000-0002-2165-6469;
JJB, 0000-0002-3222-0564; ED, 0000-0003-4129-9849;
AH, 0000-0001-9051-6876

Little is known about the structure and molecular arrangement of α- and β-amyrin, a class of triterpenoids found within the cuticle of higher plants. Blends of both amyrin isomers with different ratios have been studied taking into consideration a combined methodology of density functional theory (DFT) calculations with experimental data from scanning electron microscopy, differential scanning calorimetry and Raman vibrational spectroscopy. Results indicate that trigonal trimeric aggregations of isomer mixtures are more stable, especially in the $1:2$ $(\alpha:\beta)$ ratio. A combination of Raman spectroscopy and DFT calculations has allowed to develop an equation to determine the amount of β-amyrin in a mixed sample.

## 1. Introduction

Epidermal cells of fruits, leaves and other non-woody aerial plant organs synthesize an extracellular matrix named cuticle, which covers the outer surfaces of plants and plays an important role in the plant–environment interaction [1,2].

Cuticular waxes, together with cutin, are the lipid components of the cuticle. They can be deposited on the outer surface (epicuticular waxes) or embedded within the cuticle (intracuticular waxes) [3]. Their chemical composition is heterogeneous since they comprise a mixture of very long-chain alkanes, alcohols and fatty acids as well as triterpenoids, all of them in variable proportions [4].

Terpenes are organic compounds derived from isoprenes, with the formula $(C_5H_8)_n$. Triterpenoids, such as amyrins and the ursolic and oleanolic acids, are biologically versatile molecules based on three terpene units and a pentacyclic structure. Amyrins are located within the intracuticular wax fraction where they have

**Figure 1.** Structure of α-, β- and δ-amyrin isomers, (a), (b) and (c), respectively. Rings are named in blue.

been shown to act as nanofillers, reducing the mobility of the cutin and thus conferring mechanical strength to the cuticle [5]. Of the different amyrin isomers identified in plant cuticles (figure 1), α- and β-amyrins are the most ubiquitous and can be present as a dominant isomer like α-amyrin in Asian pear [6] or as a mixture of different isomers like in tomato fruit cuticle [7]. Nevertheless, both α- and β-amyrins are frequently found together in most of plant species [8–13].

In a previous study, we showed that isomeric pure amyrins form arranged clusters of aggregates [3], but little is known about their structure in isomeric mixed samples. However, as it has been mentioned, α- and β-amyrins are commonly present together in the plant cuticle, thus it would be important to discern whether both isomers could cluster independently or constitute stable heterogeneous structures. To answer these questions, a combination of theoretical calculations using the density functional theory (DFT) method and experimental data obtained from Raman microscopy and differential scanning calorimetry (DSC) was employed together with observations derived from scanning electron microscopy (SEM).

## 2. Methodology

### 2.1. Computational details

DFT calculations were performed with Gaussian 16 software [14] using the B3LYP functional together with the 6-31G** basis set. This is a hybrid functional combining the Hartree–Fock and Becke exact exchange functionals [15,16] with the Lee–Yang–Parr (LYP) correlation functional [17]. It has been widely employed in geometric optimizations and in the evaluation of vibration frequencies. An empirical dispersion correction GD3 was used for the analysis of long-range intermolecular interactions [18]. Structures were optimized within an *n*-octanol environment using the polarizable continuum model in order to mimic the average polarity present in the cutin matrix [19]. Theoretical Raman spectra were constructed after calculation of the vibrational normal modes using a full width at half maximum (FWHM) of $10 \, \text{cm}^{-1}$. Calculations were carried out in the Supercomputing and Bioinnovation Centre (SCBI) of the University of Málaga. The graphic editing of the optimized structures was done with the Chimera 1.11.2 software [20]. Measurements of structural parameters were performed with Mercury 3.9 software [21–23].

The relative binding energy (RBE) allows the comparison between the stabilization of each aggregate formed and the corresponding monomeric species. This parameter can be calculated as

$$\text{RBE} = \frac{E_a - n.E_m}{n},$$

where $E_a$ is the potential energy for $n$ aggregated molecules and $E_m$ is the potential energy for the isolated monomer.

Gibbs free energy of the selected trimers (ΔG) was calculated as the difference between the free energy of the trimer and the dimer-monomer set (ΔG = $G_{\text{trimer}}$ − ($G_{\text{dimer}}$ + $G_{\text{monomer}}$)). These parameters were obtained from frequency calculations in Gaussian.

### 2.2. Sample preparation

Solutions of α- and β-amyrin (Sigma-Aldrich) in CHCl$_3$ were prepared in different proportions (α : β—2 : 1, 1 : 1 and 1 : 2) with a final $4 \, \text{mg ml}^{-1}$ concentration. Samples were prepared by drop-casting method at

room temperature on silica glass substrates using aliquots of 50 µl of each solution. A total of six samples were obtained, one for each $\alpha:\beta$ proportion: $\alpha$ (pure $\alpha$), $2\alpha:\beta$ ($\alpha:\beta$—2:1), $\alpha:\beta$ ($\alpha:\beta$—1:1), $\alpha:2\beta$ ($\alpha:\beta$—1:2), $\beta$ (pure $\beta$) and the corresponding control (pure $CHCl_3$).

## 2.3. Raman spectroscopy

Raman scattering spectra were recorded with a Bruker Senterra dispersive Raman microscope equipped with a Neon lamp and using a diode laser with excitation at $\lambda = 785$ nm and 12.5 mW power at the sample point. Samples were registered at room temperature. Each spectrum was the average of 10 scans with a 4 cm$^{-1}$ spectral resolution. Raman microscopy measurements were made on three different areas for each sample.

## 2.4. Scanning electron microscopy

Samples were deposited on aluminium mounts covered with a conductive double-sided tape and gold-coated. Images were obtained with a scanning electron microscope (Jeol JSM6490 LV) at 15 kV.

## 2.5. Calorimetry: specific heat measurements

Reversible heat capacity of samples ($Cp_{rev}$) was measured at 35°C using a TA Instrument DSC Q-20 working in temperature modulated mode and under dry $N_2$ purge of 50 ml min$^{-1}$. The calorimeter was previously calibrated with a sapphire reference at a theoretical $Cp_{rev} = 0.796$ J g$^{-1}$ °C$^{-1}$. Aluminium TZero non-hermetic sample and reference crucibles, differing by less than 0.03 mg, were used. Temperature modulation was 1°C/120 s and $Cp_{rev}$ values were collected after 30 min stabilization at 35°C. Samples were analysed as prepared and after *in situ* drying in the DSC furnace at 70°C for 30 min and $Cp_{rev}$ values differed by less than 0.8%. The amount of sample used ranged between 2 and 2.5 mg and their weight was accurately measured with 0.01 mg precision.

# 3. Results and discussion

## 3.1. Scanning electron microscopy analysis

Noticeable differences between the amyrin samples crystallization were found by SEM, depending on the $\alpha:\beta$ proportion (figure 2). Pure α-amyrin crystals presented a fine needle shape with a very high length : width aspect ratio. By contrast, pure β-amyrin also showed linear structures but with a lower aspect ratio due to both the increment of width and the reduction of length of crystals. A closer look at SEM images indicated that β-amyrin crystals may be formed by the packing of needles into sheets favouring a two-dimensional growing mechanism. The lower aspect ratio structures in β-amyrin may be due to its molecular architecture. This isomer presents two methyl groups attached to the same carbon at E ring favouring the two-dimensional packing rather than the one-dimensional growth of the needle-like crystals (figure 1). As it was previously demonstrated [3], pentacyclic triterpenoids have a great tendency to assemble by non-electrostatic interactions which means molecules overlapping with no specific attraction force. A more general view and perspective can be appreciated in the electronic supplementary material, figure S1.

Samples with a blend of α- and β-amyrin ($2\alpha:\beta$, $\alpha:\beta$ and $\alpha:2\beta$) showed a gradual decrease in aspect ratio as the amount of β-amyrin increased and the structures of pure α- and β-amyrin are hardly present in the blends.

## 3.2. Optimized aggregations of $\alpha$- and $\beta$-amyrin blends

DFT calculations were carried out under the consideration that mixed α- and β-amyrins aggregate following the same structural pattern as the pure isomeric amyrins previously studied [3]. Thus, the most stable dimer conformations reported (dim3, 5 and 6, electronic supplementary material, figure S2) were employed to optimize the heterodimer. Similarly, homotrimers of α- and β-amyrin (trim31, 32, 51, 52, 61 and 62, electronic supplementary material, figure S2) were employed to optimize the heterotrimers with two different isomer ratios 2:1 (two molecules of α-amyrin with a single unit of β-amyrin, $2\alpha:\beta$) and 1:2 (one molecule of α-amyrin with two units of β-amyrin, $\alpha:2\beta$). To generate

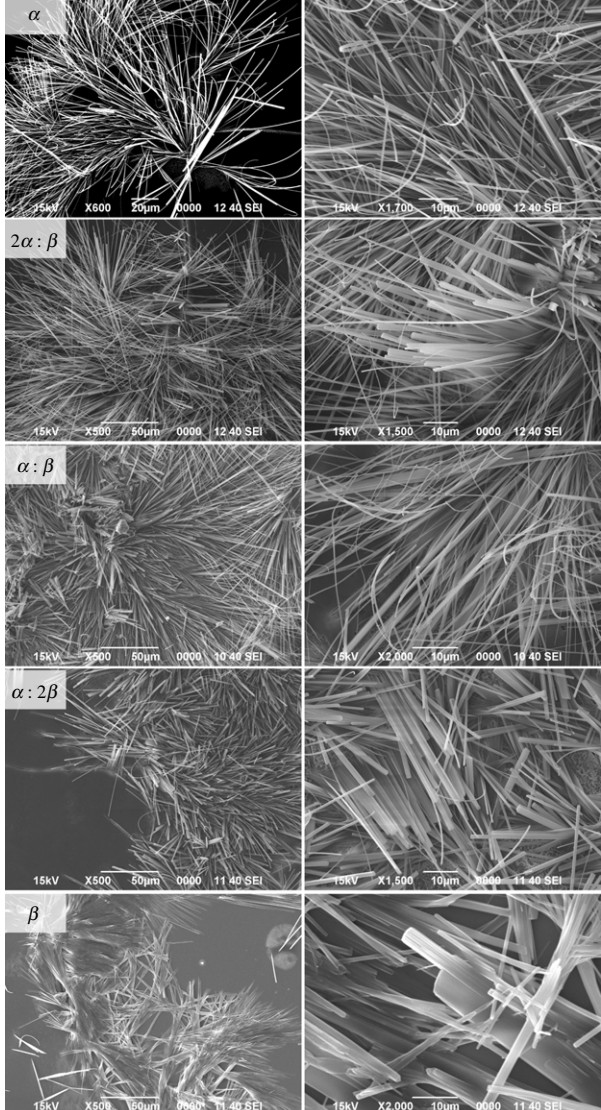

**Figure 2.** SEM images at two different magnifications (left and right columns) of different amyrin samples. Pure α-amyrin (α) β-amyrin (β) and their respective α : β mixtures: 2 : 1 (2α : β), 1 : 1 (α : β) and 1 : 2 (α : 2β).

the 2α : β and α : 2β aggregates, an α- or β- unit was added to either an α-homodimer, β-homodimer or the heterodimer. Thus, a 2α : β trimer can be αα_β-trim i.e. an α-homodimer with an additional β- unit, or αβ_α-trim, a heterodimer with an additional α- unit. A schematic representation of all these aggregates can be visualized in the electronic supplementary material, figure S2. All these possible combinations were optimized and their geometries presented in the electronic supplementary material, figure S3. As an example, figure 3 shows the homo and heterotrimers derived from dim6 and trim62 as well as the heterodimer.

A stability analysis of all the optimized structures was afforded with the calculation of their RBEs (table 1). Comparison of dimers showed that the heterodimer was energetically favoured in comparison with the α- or β-amyrin homodimers, with the exception of dim5 where the α-homodimer showed the lowest RBE.

Incorporation of a third amyrin unit to the homo or heterodimers further stabilized the aggregates, especially those with a trigonal arrangement (trim32, trim52 and trim62; table 1). This was previously reported for the homotrimers [3] and is corroborated with the heterotrimers, regardless of the isomer proportion. Notable variations in the stabilization energy were observed for the mixtures of α- and β-amyrin isomers depending on the trimer conformation, isomer proportion and whether they derive from a homo or heterodimer (table 1). Nonetheless, the trim52 conformation showed the highest stabilization for most of the blends and the most stable structure analysed was the ββ_α-trim52. With

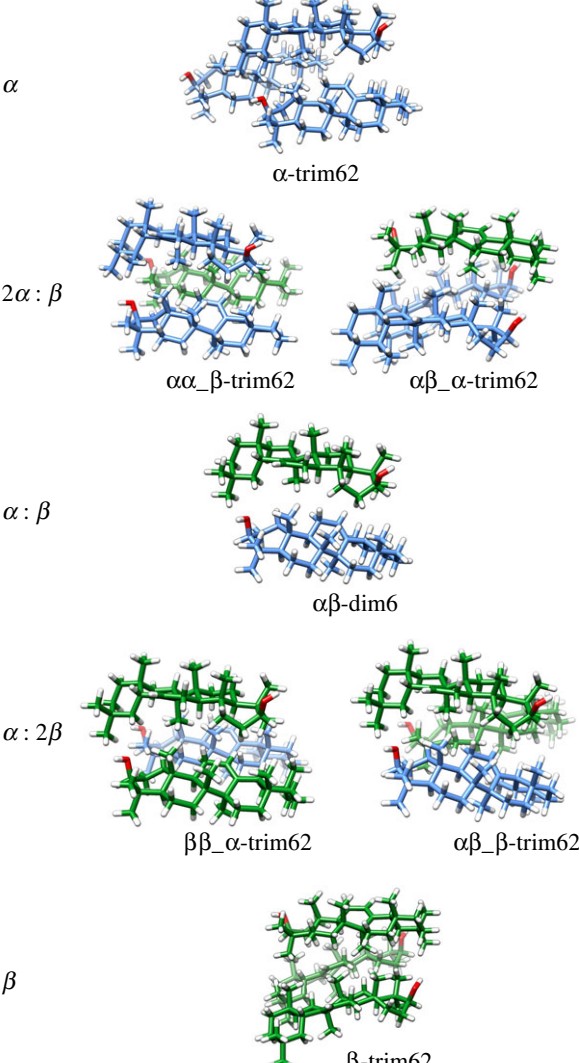

α

α-trim62

2α : β

αα_β-trim62          αβ_α-trim62

α : β

αβ-dim6

α : 2β

ββ_α-trim62          αβ_β-trim62

β

β-trim62

**Figure 3.** Optimized geometry of α- and β-amyrin homo and heterotrimers based on trim62 and the dim6 heterodimer. In blue α-amyrin units, in green β-amyrins.

**Table 1.** RBE, in kcal mol$^{-1}$, for the different proposed aggregated structures of α- and β-amyrin. Data on homodimers and homotrimers (taken from Gómez-Pulido *et al.* [3]) are shown for comparison purposes.

| RBE (kcal mol$^{-1}$) | dim3 | | dim5 | | dim6 | |
|---|---|---|---|---|---|---|
| α | −7.2 | | −8.7 | | −8.2 | |
| αβ | −7.7 | | −8.3 | | −9.3 | |
| B | −7.3 | | −6.4 | | −7.4 | |
| | trim31 | trim32 | trim51 | trim52 | trim61 | trim62 |
| α | −9.1 | −11.2 | −10.3 | −13.8 | −9.5 | −12.0 |
| αα_β | −9.3 | −10.8 | −9.1 | −13.4 | −12.7 | −12.2 |
| αβ_α | −9.2 | −11.9 | −8.9 | −13.4 | −12.1 | −13.2 |
| αβ_β | −9.0 | −10.7 | −8.9 | −12.9 | −13.0 | −13.1 |
| ββ_α | −9.3 | −11.8 | −8.5 | −14.5 | −12.3 | −13.0 |
| β | −9.2 | −13.4 | −11.0 | −12.8 | −13.5 | −13.5 |

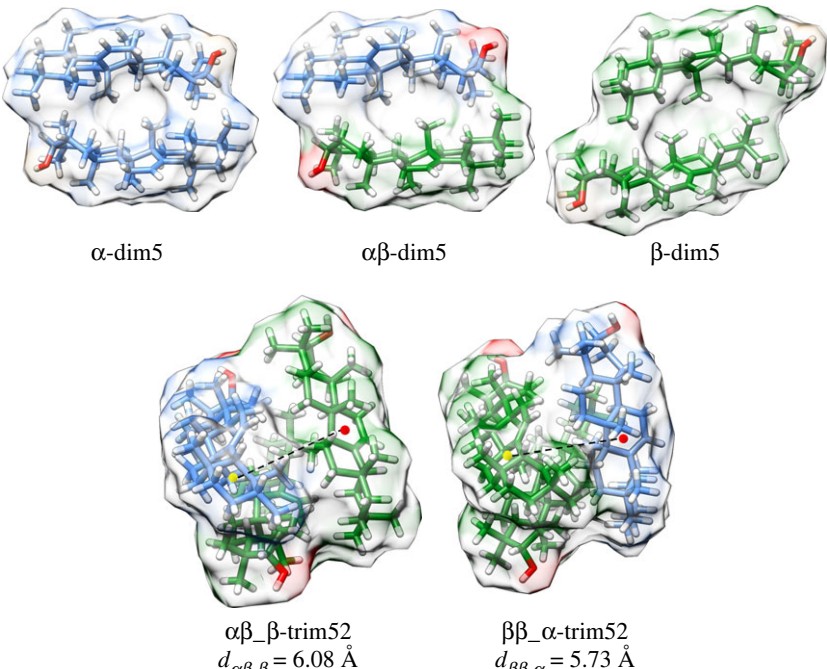

**Figure 4.** Optimized geometry, displaying the Van der Waals surfaces, of dim5 structures for amyrin homo and heterodimers (top) and two trigonal heterotrimers with a $\alpha : 2\beta$ ratio (bottom). Dashed line shows the distance between the centre of mass of the dimer (yellow circle) and the third amyrin unit (red circle).

the exception of ββ_α-trim52, none of the heterotrimers showed higher stabilization than the most stable homotrimer. Although ββ_α-trim52 and αβ_β-trim52 have the same $\alpha : \beta$ ratio (1 : 2) and an equivalent trigonal arrangement (trim52), there was a significant difference in their corresponding stabilization energy. This variation can be attributed to the relative position of the molecules between the dimer and the third unit.

Figure 4 displays the optimized molecular structure of α-dim5, αβ-dim5, β-dim5 and the trigonal trimers ββ_α-trim52 and αβ_β-trim52. An internal cavity between the monomer units can be observed in all the dimers. This cavity is able to allocate the third unit in trigonal trimers (figure 4 top). This could explain the higher stability of the trigonal trimers (trim32, trim52 and trim62) with respect to the in-block trimers (trim31, trim51 and trim61).

Several structural parameters were analysed in the homo and hetero-dim5 structures (electronic supplementary material, table S1). Only small differences were found between α-dim5 and αβ-dim5, the relative shift between the units ($\theta_s$) and distance between opposed rings of different units ($d_r$) were very similar and the α-amyrin units of both dimers barely changed the total bending of the molecular structure ($\theta_b$) or distortion of the E ring with respect to the molecular backbone ($\theta_E$). However, β-dim5 presented a lower stabilization (table 1) and higher molecular separation and shifting between units. Thus, both β-amyrin molecules in β-dim5 showed a more distorted structure compared to the β-amyrin unit in αβ-dim5. This could be explained by the steric hindrance that the two methyl groups attached to the same carbon at the E ring in β-amyrin produce between both molecules in β-dim5.

Distance between the centre of mass of the dimer and the third amyrin unit was lower for ββ_α-trim52 than αβ_β-trim52 (figure 4). This indicates that β-dim5 allows a better stacking of the α-amyrin third unit, thus conferring a better stabilization to ββ_α-trim52, as shown by its lowest RBE (table 1). Free energy changes after the incorporation of a third unit to the homo and heterodimer were calculated for both trimers. The energies obtained were $\Delta G_{\beta\beta\_\alpha} = -46.78$ kJ mol$^{-1}$ and $\Delta G_{\alpha\beta\_\beta} = -21.73$ kJ mol$^{-1}$ and confirmed the results of the previous analyses. Thus, aggregation of an α- unit to the β-homodimer was more energetically favoured than the incorporation of a β-unit to the heterodimer.

## 3.3. Calorimetry of the amyrin blends

It is well established that the specific heat (or heat capacity), $Cp$, of a molecule is the most sensitive thermodynamic indicator of structure. Depending on the nature of their individual components, the

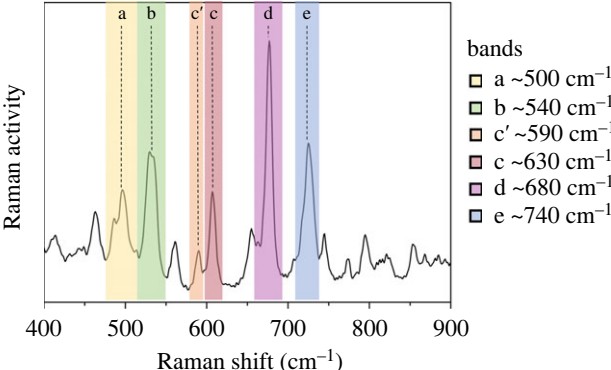

**Figure 5.** Experimental Raman spectrum of the $\alpha : \beta$ 1 : 1 sample ($\alpha : \beta$) showing the selected C-C-C scissoring bands and the c′ shoulder.

thermodynamic behaviour of a solid–solid solution often shows pronounced non-ideality owing to the formation of intermolecular interactions. In this sense, the named excess heat capacity gives useful structural information on the behaviour of the corresponding solute–solute interactions in a solid solution [24,25].

Measured specific heat values of $\alpha$- and β-amyrin at 35°C (308 K) were 1.32 and 1.16 J g$^{-1}$ K$^{-1}$, respectively. These values are of the same order of magnitude as other chemical components of plant waxes [24,25]. On the other hand, specific heat values for the $2\alpha : \beta$, $\alpha : \beta$ and $\alpha : 2\beta$ mixtures were 0.86, 0.97 and 1.13 J g$^{-1}$ K$^{-1}$, respectively. Taking into consideration the different isomer ratios of the mixtures, it is clear that none of them has a positive excess specific heat. This is an indication that the mixtures are far from an ideal solid–solid solution. Besides, negative excess $Cp$ is an indication that there is a decrease in structure of the short-range orientational order. Thus, the molecular order of one isomer amyrin molecule is partially destroyed by the presence of a molecule of the other isomer and consequently the self-association of the pentacyclic structures conduce to the formation of a less crystallinity structure. The differences between amyrin isomers lead to steric hindrances and disruption of the geometric homogeneity of the respective isomeric structures. Nevertheless, it is noticeable that the $\alpha : 2\beta$ mixture presented the lowest negative excess heat capacity in good agreement with its high stability (table 1).

## 3.4. Raman spectroscopy of the amyrin blends

Raman spectra within the 500–750 cm$^{-1}$ range were registered on three different microscopic areas for each pure isomer sample and the isomeric mixtures (electronic supplementary material, figure S4). The relative intensity of the five bands (bands a–e in figure 5) associated with the C-C-C scissoring of the amyrin backbone were measured for each sample and area and normalized with the 680 cm$^{-1}$ band (band d), following the approach already published [3]. Raman relative intensities ($I_R$) were then calculated to determine the differences among the different experimental spectra for each isomer and blend sample (electronic supplementary material, figure S5). Spectra of α- and β-amyrin in the three areas registered were very similar, indicating that pure amyrins have a more homogeneous distribution. However, isomer blends showed differences in the $I_R$ pattern among the different areas analysed, suggesting a higher degree of variability within the samples. This could be due to the presence of a variable array of aggregates within the sample and it would be in agreement with the heterogeneous distribution of different crystal structures observed in SEM for the mixtures.

A detailed analysis of the experimental Raman spectra showed notable intensity variation of the band shoulder c′ at approximately 590 cm$^{-1}$ between α- and β-amyrin samples. This c′ band shoulder was associated with the C-C-C scissoring vibration in the E ring of the amyrin backbone i.e. the ring where the methyl groups that differentiate α- and β-amyrin are located. Normalization of the intensity of this c′ shoulder with the band c was carried out and the c′/c values were 0.08 for α-amyrin and 0.46 for β-amyrin. Isomer mixtures displayed intermediate values but with higher c′/c ratios as the proportion of β-amyrin in the blend increased from 33 to 67% (0.29 for $2\alpha : \beta$, 0.38 for $\alpha : \beta$ and 0.45 for $\alpha : 2\beta$). This parameter was calculated for all the pure isomeric amyrin conformers (electronic supplementary material, table S2). The eigenvector for the c′ band shoulder for β-amyrin monomer is presented as well as in the electronic supplementary material, figure S6. These data prove that the higher intensity of the c′ band is associated with the position of the methyl groups in E ring.

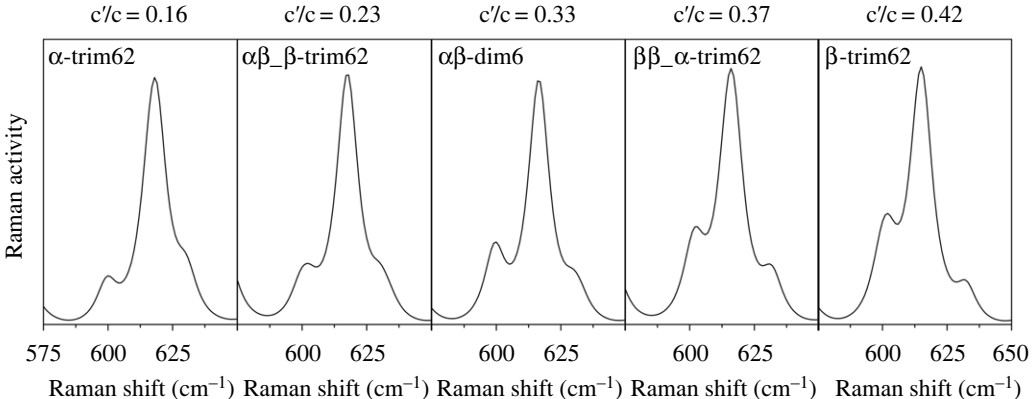

**Figure 6.** Normalized theoretical Raman spectra of the trim62 homo and heterotrimers and dim6 heterodimer. The 575–650 cm$^{-1}$ spectral range that comprises the c' and c bands is shown. The corresponding c'/c ratio is indicated on the top of each spectrum.

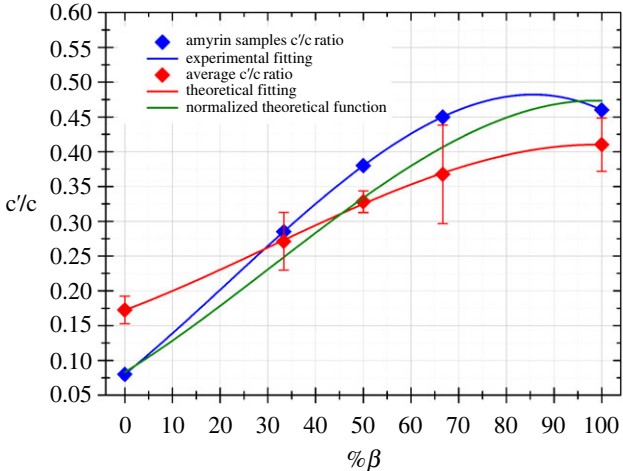

**Figure 7.** Graph of c'/c versus percentage of β-amyrin in the structure/sample for calculated (red) and experimental (blue) spectra. Standard deviations for the average calculated data are also shown. The fitting functions (1) and (3) are represented as a red and blue lines, respectively. The adjusted function (2) is also represented in green line.

A theoretical Raman and the corresponding c'/c ratio were calculated for each amyrin isomer, homo and heterodimers as well as the homo and heterotrimers with different isomer ratios (electronic supplementary material, figure S7). Most of aggregates showed the same trend, with the c'/c ratio increasing with the amount of β-isomer in the structure. As an example, figure 6 shows the c'-c region of the Raman theoretical spectra for trim62 homo and heterotrimers and dim6 heterodimer.

In figure 7, a plot of the theoretical c'/c ratio versus the percentage of $\beta$ is represented. The curve could be adjusted to a cubic function (red line) with a good fitting ($R^2 = 0.998$):

$$c'/c = 0.17 + 3.02 \times 10^{(-3)} \cdot \%\beta + 8.48 \times 10^{(-6)} \cdot (\%\beta)^2 - 1.55 \times 10^{(-7)} \cdot (\%\beta)^3. \tag{3.1}$$

The curve and equation indicate that the presence of β-amyrin isomer is progressively manifested, from a spectroscopic point of view, as it is added to an α-amyrin sample. This agrees with the SEM observations mentioned above.

Equation (3.1) was mathematically recalculated in order to adjust it to the experimental data. For this purpose, the function was normalized with the c'/c data from the experimental Raman of both isomer samples (c'/c = 0.08 for an α-amyrin sample with 0%β and c'/c = 0.46 for a 100% β-amyrin sample). Thus, the curve fitting could be applicable to any c'/c ratio obtained from the experimental spectra. The resulting equation was

$$c'/c = 0.08 + 4.96 \times 10^{(-3)} \cdot \%\beta + 1.39 \times 10^{(-5)} \cdot (\%\beta)^2 - 2.55 \times 10^{(-7)} \cdot (\%\beta)^3. \tag{3.2}$$

**Table 2.** c′/c ratio determined from experimental Raman spectra registered at three different areas for each amyrin blend. Percentage of $\beta$-amyrin calculated from the Raman spectra following equation (3.2) is also included.

| sample | area1 | | area2 | | area3 | |
|---|---|---|---|---|---|---|
| | c′/c | %$\beta$ | c′/c | %$\beta$ | c′/c | %$\beta$ |
| $2\alpha:\beta$ | 0.23 | 30.01 | 0.31 | 37.90 | 0.26 | 30.00 |
| $\alpha:\beta$ | 0.40 | 53.67 | 0.37 | 48.84 | 0.36 | 45.83 |
| $\alpha:2\beta$ | 0.45 | 66.48 | 0.42 | 58.82 | 0.40 | 68.30 |

This function is plotted in figure 7 (green line) and was employed to determine the percentage of β-amyrin in each of the blends from their experimental Raman spectra (table 2). Comparison of the results obtained for each area within the same blend showed a certain degree of variability, indicating a heterogeneous distribution of the different isomers within a sample. However, calculated percentages of β-amyrin were well within the expected ranges of 33.3% in $2\alpha:\beta$, 50% in $\alpha:\beta$ and 66.7% in $\alpha:2\beta$.

These results demonstrate that there is a good correlation between the experimental and theoretical data and that these data can be fitted to a cubic function. Experimental data were then adjusted to this type of function (see blue line in figure 7) and equation (3.3) was obtained ($R^2 = 1$).

$$c'/c = 0.08 + 5.61 \times 10^{(-3)} \cdot \%\beta + 3.36 \times 10^{(-5)} \cdot (\%\beta)^2 - 5.17 \times 10^{(-7)} \cdot (\%\beta)^3. \tag{3.3}$$

This equation can be used to relate the c′/c ratio of a mixture of amyrin isomers with the percentage of β-amyrin present.

## 4. Conclusion

DFT optimization of isolated and blends of both isomers gives a theoretical approximation on the stability of the aggregates depending on isomer ratio and geometrical arrangement. Steric hindrance of the two methyl groups of β-amyrin favoured trigonal heterotrimeric aggregations. For any amyrin mixture, there is a heterogeneous distribution of both isomers within a sample that can be determined by a combination of Raman spectroscopy and DFT calculations. A band ratio (c′/c) has been shown to be indicative of the percentage of β-amyrin and an equation has been developed to estimate the percentage of β-amyrin in an isomer mixture from its Raman spectra.

Calorimetric data showed a decrease in the crystalline structure of the blends in comparison with the pure isomers that agreed with the theoretical calculations and supports the differences in length : width aspect ratio observed using SEM between α- and β-amyrin samples and in the blends.

These results provide valuable information on the arrangement of amyrin isomers in different *in vitro* scenarios. In addition, the present research will allow a better understanding of the location and molecular arrangement of these triterpenoids within the plant cuticle.

Data accessibility. Additional information concerning this paper is available in the electronic supplementary material and in Dryad Digital Repository: https://datadryad.org/stash/share/4-47x80yRF6BnboKY8FaFwOMRjCpsHAethLOJ8roug0.
    The data are provided in the electronic supplementary material [26].
Authors' contributions. L.D.M.G.-P.: data curation, investigation, methodology, writing—original draft and writing—review and editing; R.C.G.-C.: conceptualization, investigation, supervision, validation, writing—original draft and writing—review and editing; J.J.B..: data curation, resources and writing—original draft; E.D.: funding acquisition, project administration, supervision, validation, writing—original draft and writing—review and editing; A.H.: funding acquisition, project administration, supervision, validation, writing—original draft and writing—review and editing.
    All authors gave final approval for publication and agreed to be held accountable for the work performed therein.
Conflict of interest declaration. There are no conflicts to declare.
Funding. This work was supported by grant no. RTI2018-094277-B/AEI/10.13039/501100011033 from Agencia Estatal de Investigación, Ministerio de Ciencia e Innovación, Spain, co-financed by the European Regional Development Fund (ERDF). L.D.M.G.-P. is the recipient of a FPI fellowship (BES-2016-078716) from Spanish MINECO co-funded by the European Social Fund.
Acknowledgements. The authors thankfully acknowledge the computer resources, technical expertise and assistance provided by the SCBI (Supercomputing and Bioinformatics) centre and Servicios Centrales de Apoyo a la Investigación (SCAI) of the University of Málaga.

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
