## [Peer Review File · Royal Society Open Science]

Review History

RSOS-211238.R0 (Original submission)

Review form: Reviewer 1

Is the manuscript scientifically sound in its present form?

No

Are the interpretations and conclusions justified by the results?

No

Is the language acceptable?

Yes

Do you have any ethical concerns with this paper?

No

Have you any concerns about statistical analyses in this paper?

Yes

Recommendation?

Reject

Comments to the Author(s)

The authors have published an article recently in RSC Advances entitled "Structure determination of amyrrin isomers in" cuticular waxes: a combined DFT/vibrational spectroscopy methodology". This manuscript is an incremental work of their previous publication. Hence, it is not recommended for publication.

Comments:

1. In section 3.1 Scanning Electronic Microscopy analysis:

The authors claim "Remarkable differences in the crystal structure of both isomers was observed", but the no such remarkable differences in SEM images was observed. Moreover, The Scanning Electronic Microscopy images depends on the method of sample preparation, incubation time, nature of the solvent (polar/ nonpolar) etc.

2. In section 3.1 Scanning Electronic Microscopy analysis:

The authors also claim "Mixtures of both isomers ($2\alpha:\beta$, $\alpha:\beta$ and $\alpha:2\beta$) showed an intermediate behavior between them, with crystallization depending on the $\alpha:\beta$ proportion", but the statement is inconsistent with the SEM images. How did the authors predict the stability of the intermediates?

3. In section 3.1 Scanning Electronic Microscopy analysis:

The authors claims "Thus samples rich in β -amyrrin form less ordered aggregates than the α -amyrrin". But SEM analysis does not support the statement adequately.

4. In section 4. Conclusion: The authors stated "Scanning Electron Microscopy showed microscopic differences between α - and β - amyrrins, with the former displaying long and thin needle crystal structures and the latter a combination of film-like structure and shorter needles of variable width. Isomer mixtures showed variable arrangements of both isomers instead of separated α - and β -amyrrin structural domains." But no such remarkable difference was observed from section 3.1 Scanning Electronic Microscopy analysis.

Review form: Reviewer 2 (Ravi Kumar Venkatraman)

Is the manuscript scientifically sound in its present form?

Yes

Are the interpretations and conclusions justified by the results?

Yes

Is the language acceptable?

Yes

Do you have any ethical concerns with this paper?

No

Have you any concerns about statistical analyses in this paper?

No

Recommendation?

Major revision is needed (please make suggestions in comments)

Comments to the Author(s)

Reviewer comments for the manuscript ID RSOS-211238:

Manuscript Title: Structural analysis of mixed α - and β -amyrin samples

In this manuscript, the authors have studied the composite structures of α - and β - isomers of amyryns, a triterpenoid that is ubiquitously present in the epidermal cells of plants. To this end, the authors have utilized scanning electron microscopy to study the crystal structure of pristine and a mixture of amyrin isomers. In addition, the authors have used calorimetry and Raman spectroscopy methods to establish that the composite structures are formed with amyrin isomers rather than the isolated mixtures of pure isomers. Furthermore, the authors have also performed the DFT calculations studies to corroborate their experimental results.

This manuscript is well-written and nicely presented. Also, the manuscript falls within the aim and scope of Royal Society Open Science. However, a few major and minor concerns are listed below. Therefore I recommend this manuscript for publication after a Major Revision, and I would like to be invited to review the revised manuscript.

Major Concerns

1.) The authors claim that the SEM images of α - β - isomers amyryns and their mixture shows a distinct crystal structure. For example, Figure 2 second image from the top (left) displays a magnified SEM image for $2\alpha:\beta$ amyryns. It appears like there are two isolated crystal structures, for example, one at the center with short and wider needle-like structure characteristic of β amyrin surrounded by long thin needle-like structure characteristics of α -amyrin. The authors must clarify how these SEM images confirm that the mixture of isomers of amyrin forms a composite structure which is not evident from the SEM images. The authors have to rephrase that the more supporting evidence arises from calorimetry and Raman studies and must be discussed for the reader's clarity.

2.) In section 3.2, the origin of dim3, trim31, etc., must be explained in detail for the reader's brevity.

3.) The authors describe that the ratio of band c' to c was chosen to evaluate the composition of different isomers in the mixtures of α -, and β -amyryns. It would be nice if the authors could provide statistical analysis like singular value decompositions and principal component analysis of the Raman spectra of α -, and β -amyryns and prove that the band c' indeed shows large variation with change in the composition of isomers.

Minor Concerns

1.) The authors must briefly introduce the structure of terpenoids and triterpenoids in the Introduction section for the reader's clarity.

2.) In the Methodology section (2.3. Raman spectroscopy), the authors have described the Raman excitation wavelength used in this study as 785 nm, which is generated using Nd:YAG laser. But the fundamental of Nd:YAG laser and its second harmonics emit at ~ 1064 and 532 nm wavelengths, respectively. Therefore, the authors must clarify whether they have used a 785 nm diode laser or second harmonics of Nd:YAG laser (532 nm) as Raman excitation wavelength.

3.) The authors mention (Results and Discussion section, Page no. 2) that non-electronic interactions drive the assembly of homo-/hetero-oligomers of amyryns. However, it's not clear whether the author means non-electrostatic or hydrophobic interactions. It must be clarified.

My decision: Major Revision

Decision letter (RSOS-211238.R0)

Dear Dr González-Cano:

Manuscript ID: RSOS-211238

Title: "Structural analysis of mixed α - and β -amylin samples"

Thank you for submitting the above manuscript to Royal Society Open Science. Your paper was sent to reviewers and their comments are included at the bottom of this letter.

In view of the concerns raised by the reviewers, the manuscript has been rejected in its current form. However, a new manuscript may be submitted which takes into consideration these comments.

Please note that resubmitting your manuscript does not guarantee eventual acceptance, and that your resubmission will be subject to peer review before a decision is made.

Your resubmitted manuscript should be submitted by 04-May-2022. If you are unable to submit by this date please contact the Editorial Office.

Yours sincerely,
Dr Ellis Wilde
Publishing Editor, Journals

On behalf of the Subject Editor Professor Anthony Stace and the Associate Editor Dr Debashree Ghosh

REVIEWER(S) REPORTS:
Associate Editor Comments to Author ():
RSC Associate Editor

Comments to the Author:

There are major reservations expressed by the referees and therefore, the manuscript cannot be accepted at this point. However, the authors may address the issues raised by the referees and resubmit with a point-by-point reply.

RSC Subject Editor

Comments to the Author:

(There are no comments.)

Reviewers' Comments to Author:

Reviewer: 1

Comments to the Author(s)

The authors have published an article recently in RSC Advances entitled "Structure determination of amyirin isomers in" cuticular waxes: a combined DFT/vibrational spectroscopy methodology". This manuscript is an incremental work of their previous publication. Hence, it is not recommended for publication.

Comments:

1. In section 3.1 Scanning Electronic Microscopy analysis:

The authors claim "Remarkable differences in the crystal structure of both isomers was observed", but the no such remarkable differences in SEM images was observed. Moreover, The Scanning Electronic Microscopy images depends on the method of sample preparation, incubation time, nature of the solvent (polar/ nonpolar) etc.

2. In section 3.1 Scanning Electronic Microscopy analysis:

The authors also claim "Mixtures of both isomers ($2\alpha:\beta$, $\alpha:\beta$ and $\alpha:2\beta$) showed an intermediate behavior between them, with crystallization depending on the $\alpha:\beta$ proportion", but the statement is inconsistent with the SEM images. How did the authors predict the stability of the intermediates?

3. In section 3.1 Scanning Electronic Microscopy analysis:

The authors claims "Thus samples rich in β -amyirin form less ordered aggregates than the α -amyirin". But SEM analysis does not support the statement adequately.

4. In section 4. Conclusion: The authors stated "Scanning Electron Microscopy showed microscopic differences between α - and β - amyirins, with the former displaying long and thin needle crystal structures and the latter a combination of film-like structure and shorter needles of variable width. Isomer mixtures showed variable arrangements of both isomers instead of separated α - and β -amyirin structural domains." But no such remarkable difference was observed from section 3.1 Scanning Electronic Microscopy analysis.

Reviewer: 2

Comments to the Author(s)

Reviewer comments for the manuscript ID RSOS-211238:

Manuscript Title: Structural analysis of mixed α - and β -amyirin samples

In this manuscript, the authors have studied the composite structures of α - and β - isomers of amyirins, a triterpenoid that is ubiquitously present in the epidermal cells of plants. To this end, the authors have utilized scanning electron microscopy to study the crystal structure of pristine and a mixture of amyirin isomers. In addition, the authors have used calorimetry and Raman spectroscopy methods to establish that the composite structures are formed with amyirin isomers

rather than the isolated mixtures of pure isomers. Furthermore, the authors have also performed the DFT calculations studies to corroborate their experimental results.

This manuscript is well-written and nicely presented. Also, the manuscript falls within the aim and scope of Royal Society Open Science. However, a few major and minor concerns are listed below. Therefore I recommend this manuscript for publication after a Major Revision, and I would like to be invited to review the revised manuscript.

Major Concerns

1.) The authors claim that the SEM images of α - β - isomers amyryns and their mixture shows a distinct crystal structure. For example, Figure 2 second image from the top (left) displays a magnified SEM image for $2\alpha:\beta$ amyryns. It appears like there are two isolated crystal structures, for example, one at the center with short and wider needle-like structure characteristic of β amyryn surrounded by long thin needle-like structure characteristics of α -amyryn. The authors must clarify how these SEM images confirm that the mixture of isomers of amyryn forms a composite structure which is not evident from the SEM images. The authors have to rephrase that the more supporting evidence arises from calorimetry and Raman studies and must be discussed for the reader's clarity.

2.) In section 3.2, the origin of dim3, trim31, etc., must be explained in detail for the reader's brevity.

3.) The authors describe that the ratio of band c' to c was chosen to evaluate the composition of different isomers in the mixtures of α -, and β -amyryns.

It would be nice if the authors could provide statistical analysis like singular value decompositions and principal component analysis of the Raman spectra of α -, and β -amyryns and prove that the band c' indeed shows large variation with change in the composition of isomers.

Minor Concerns

1.) The authors must briefly introduce the structure of terpenoids and triterpenoids in the Introduction section for the reader's clarity.

2.) In the Methodology section (2.3. Raman spectroscopy), the authors have described the Raman excitation wavelength used in this study as 785 nm, which is generated using Nd:YAG laser. But the fundamental of Nd:YAG laser and its second harmonics emit at ~ 1064 and 532 nm wavelengths, respectively. Therefore, the authors must clarify whether they have used a 785 nm diode laser or second harmonics of Nd:YAG laser (532 nm) as Raman excitation wavelength.

3.) The authors mention (Results and Discussion section, Page no. 2) that non-electronic interactions drive the assembly of homo-/hetero-oligomers of amyryns. However, it's not clear whether the author means non-electrostatic or hydrophobic interactions. It must be clarified.

My decision: Major Revision

Author's Response to Decision Letter for (RSOS-211238.R0)

See Appendix A.

RSOS-211787.R0

Review form: Reviewer 2 (Ravi Kumar Venkatraman)

Is the manuscript scientifically sound in its present form?

Yes

Are the interpretations and conclusions justified by the results?

Yes

Is the language acceptable?

Yes

Do you have any ethical concerns with this paper?

No

Have you any concerns about statistical analyses in this paper?

No

Recommendation?

Accept as is

Comments to the Author(s)

The authors have answered all the reviewers' comments and incorporated all the necessary modifications in the revised manuscript. Therefore, I recommend this manuscript for publication as it is.

Review form: Reviewer 3

Is the manuscript scientifically sound in its present form?

Yes

Are the interpretations and conclusions justified by the results?

Yes

Is the language acceptable?

Yes

Do you have any ethical concerns with this paper?

No

Have you any concerns about statistical analyses in this paper?

No

Recommendation?

Accept with minor revision (please list in comments)

Comments to the Author(s)

After reading the manuscript, the comments, of the reviewers, and the replies, I feel that the authors have responded adequately to most of the reviewer comments. The computational parts of the manuscript are for the most part clearly explained and technically sound. I believe this work is publishable in Royal Society Open Science, after a few minor suggestions:

1. The reviewers mention in their reply to the reviewers that that clarified "non-electronic interactions" to mean "non-electrostatic," but it remains "non-electronic" in the manuscript.

2. It appears from the equation for RBE that it is computed from the potential energies of the dimers and the trimers compared to the monomers. Those energies are reported in Table 1 in kcal/mol. I therefore found it confusing when in page 4 there is suddenly a discussion of free energy changes reported in kJ/mol. Are those really computed free energies (e.g., based on frequency calculations)? Are RBEs in table 1 therefore all free energies or potential energies? This is somewhat unclear.

3. For binding energy calculations, it is usually important to account for basis set superposition error. However, in these cases, it could be possible to assume that BSSE would be similar for the different interactions and since the focus is on relative binding energies.

4. I recommend including the xyz coordinates of key (e.g., lowest energy) optimized structures in the supporting information for the sake of reproducibility, along with their absolute energies.

Decision letter (RSOS-211787.R0)

Dear Dr González-Cano:

Title: Structural analysis of mixed α - and β -amylin samples

Manuscript ID: RSOS-211787

Thank you for submitting the above manuscript to Royal Society Open Science. On behalf of the Editors and the Royal Society of Chemistry, I am pleased to inform you that your manuscript will be accepted for publication in Royal Society Open Science subject to minor revision in accordance with the referee suggestions. Please find the reviewers' comments at the end of this email.

The reviewers and handling editors have recommended publication, but also suggest some minor revisions to your manuscript. Therefore, I invite you to respond to the comments and revise your manuscript.

Please also include the following statements alongside the other end statements. As we cannot publish your manuscript without these end statements included, if you feel that a given heading is not relevant to your paper, please nevertheless include the heading and explicitly state that it is not relevant to your work. We have included a screenshot example of the end statements for reference.

- Ethics statement

Please clarify whether you received ethical approval from a local ethics committee to carry out your study. If so please include details of this, including the name of the committee that gave consent in a Research Ethics section after your main text. Please also clarify whether you received informed consent for the participants to participate in the study and state this in your Research Ethics section.

OR

Please clarify whether you obtained the necessary licences and approvals from your institutional animal ethics committee before conducting your research. Please provide details of these licences and approvals in an Animal Ethics section after your main text.

OR

Please clarify whether you obtained the appropriate permissions and licences to conduct the fieldwork detailed in your study. Please provide details of these in your methods section.

- Data accessibility

It is a condition of publication that you make available the data and research materials supporting the results in the article. Datasets should be deposited in an appropriate publicly available repository and details of the associated accession number, link or DOI to the datasets must be included in the Data Accessibility section of the article (<https://royalsocietypublishing.org/rsos/for-authors#question17>). Reference(s) to datasets should also be included in the reference list of the article with DOIs (where available).

Please include a Data Availability section after your main text stating where supporting data are available from, or where they will be made available should your article be accepted for publication.

If you wish to submit your supporting data or code to Dryad (<http://datadryad.org/>), or modify your current submission to dryad, please use the following link:
<http://datadryad.org/submit?journalID=RSOS&manu=RSOS-211787>

- Competing interests

Please include a Competing Interests section after your main text declaring any financial or non-financial competing interests. If you have no competing interests please state 'I/we have no competing interests.'

- Authors' contributions

Please include an Authors' Contributions section at the end of your main text detailing the contribution of each author. All authors should have read and approved the manuscript before submission and this should be stated in the Authors' Contributions section.

The list of Authors should meet all of the following criteria; 1) substantial contributions to conception and design, or acquisition of data, or analysis and interpretation of data; 2) drafting the article or revising it critically for important intellectual content; and 3) final approval of the version to be published.

- Acknowledgements

- Funding statement

Please include a funding section after your main text which lists the source of funding for each author.

Because the schedule for publication is very tight, it is a condition of publication that you submit the revised version of your manuscript before 26-Feb-2022. Please note that the revision deadline will expire at 00.00am on this date. If you do not think you will be able to meet this date please let me know immediately.

Kind regards,
Ellis Wilde
Assistant Editor, Journals

Royal Society of Chemistry
Thomas Graham House
Science Park, Milton Road

Cambridge, CB4 0WF
Royal Society Open Science - Chemistry Editorial Office

On behalf of the Subject Editor Professor Anthony Stace and the Associate Editor Dr Debashree Ghosh.

RSC Associate Editor

Comments to the Author:

The manuscript maybe accepted after the recommendations of the Referees are taken into consideration. A point by point reply of the reviewer comments and changes made should be submitted.

Reviewer comments to Author:

Reviewer: 2

Comments to the Author(s)

The authors have answered all the reviewers' comments and incorporated all the necessary modifications in the revised manuscript. Therefore, I recommend this manuscript for publication as it is.

Reviewer: 3

Comments to the Author(s)

After reading the manuscript, the comments, of the reviewers, and the replies, I feel that the authors have responded adequately to most of the reviewer comments. The computational parts of the manuscript are for the most part clearly explained and technically sound. I believe this work is publishable in Royal Society Open Science, after a few minor suggestions:

1. The reviewers mention in their reply to the reviewers that that clarified "non-electronic interactions" to mean "non-electrostatic," but it remains "non-electronic" in the manuscript.
2. It appears from the equation for RBE that it is computed from the potential energies of the dimers and the trimers compared to the monomers. Those energies are reported in Table 1 in kcal/mol. I therefore found it confusing when in page 4 there is suddenly a discussion of free energy changes reported in kJ/mol. Are those really computed free energies (e.g., based on frequency calculations)? Are RBEs in table 1 therefore all free energies or potential energies? This is somewhat unclear.
3. For binding energy calculations, it is usually important to account for basis set superposition error. However, in these cases, it could be possible to assume that BSSE would be similar for the different interactions and since the focus is on relative binding energies.
4. I recommend including the xyz coordinates of key (e.g., lowest energy) optimized structures in the supporting information for the sake of reproducibility, along with their absolute energies.

Author's Response to Decision Letter for (RSOS-211787.R0)

See Appendix B.

Decision letter (RSOS-211787.R1)

Dear Dr González-Cano:

Title: Structural analysis of mixed α - and β -amyrin samples
Manuscript ID: RSOS-211787.R1

It is a pleasure to accept your manuscript in its current form for publication in Royal Society Open Science. The chemistry content of Royal Society Open Science is published in collaboration with the Royal Society of Chemistry.

Yours sincerely,
Ellis Wilde
Publishing Editor, Journals

On behalf of the Subject Editor Professor Anthony Stace and the Associate Editor Dr Debashree Ghosh.

RSC Associate Editor
Comments to the Author:

The authors have incorporated all the changes suggested by the referees satisfactorily and therefore, the manuscript may be accepted for publication.

Reviewer(s)' Comments to Author:

Appendix A

Reviewer 1

Q1 - The authors claim "Remarkable differences in the crystal structure of both isomers was observed", but the no such remarkable differences in SEM images was observed. Moreover, The Scanning Electronic Microscopy images depends on the method of sample preparation, incubation time, nature of the solvent (polar/ nonpolar) etc.

The text has been updated in order to avoid confusion about the conclusions of the SEM technique. These data supposed a qualitative starting point for the whole investigation including Raman spectroscopy, calorimetry and DFT calculations. On other hand, all the samples were prepared and measured under the same exact conditions thus changes observed may be attributed only to isomers proportion. Literature on this point indicated that most of the recrystallization of most of the cuticle waxes present the same morphology than *in vivo* plant samples (Jeffree et al., New Phytologist, 1975, 539-549).

Q2 - The authors also claim "Mixtures of both isomers (2 α : β , α : β and α :2 β) showed an intermediate behavior between them, with crystallization depending on the α : β proportion", but the statement is inconsistent with the SEM images. How did the authors predict the stability of the intermediates?

The text has been updated to explain that the intermediate behaviour made reference to a blend of α and β structures instead of separated domains of the isomers along the sample. Considering the qualitative character of the microscopy, stability cannot be predicted by this technique, whereas it is treated in further sections of this publications with more appropriated methods.

Q3 - The authors claims "Thus samples rich in β -amyirin form less ordered aggregates than the α -amyirin". But SEM analysis does not support the statement adequately.

This statement is a conclusion of the molecular structure observed in β -amyirin which present two methyl groups attached to the same carbon at E ring. This fact and its consequences are analysed in section 3.2: "Optimized aggregations of α and β amyirin blends".

Q4 - The authors stated "Scanning Electron Microscopy showed microscopic differences between α - and β - amyirins, with the former displaying long and thin needle crystal structures and the latter a combination of film-like structure and shorter needles of variable width. Isomer mixtures showed variable arrangements of both isomers instead of separated α - and β -amyirin structural domains."

The conclusions have been updated to fix with the changes added in the microscopy section.

Reviewer 2

Major revisions

Q1 - The authors claim that the SEM images of α -, β - isomers amyryns and their mixture shows a distinct crystal structure. For example, Figure 2 second image from the top (left) displays a magnified SEM image for 2 α : β amyryns. It appears like there are two isolated crystal structures, for example, one at the center with short and wider needle-like structure characteristic of β amyryn surrounded by long thin needle-like structure characteristics of α -amyryn. The authors must clarify how these SEM images confirm that the mixture of isomers of amyryn forms a composite structure which is not evident from the SEM images. The authors have to rephrase that the more supporting evidence arises from calorimetry and Raman studies and must be discussed for the reader's clarity.

The text has been updated to explain that, as the reviewer indicated, the intermediate behaviour made reference to a blend of α and β structures instead of separated domains of the isomers along the sample.

Q2 - In section 3.2, the origin of $dim3$, $trim31$, etc., must be explained in detail for the reader's brevity.

An adequate explanation extracted from the referenced article has been added to the Supplementary Information to clarify the structures studied along the theoretical study.

Q3 - The authors describe that the ratio of band c' to c was chosen to evaluate the composition of different isomers in the mixtures of α -, and β -amyryns.

It would be nice if the authors could provide statistical analysis like singular value decompositions and principal component analysis of the Raman spectra of α -, and β -amyryns and prove that the band c' indeed shows large variation with change in the composition of isomers.

Considering the reviewer recommendations, an additional study of the c'/c ratio for the isomeric pure conformations (Table S2 in ESM) has been added to the eigenvectors analysis (Figure S6 in ESM) in order to demonstrate that this parameter is mainly dependent on the structural differences, in the E ring, between α - and β -amyryn.

We have also improved Figure 7 plot in the manuscript with the addition of the standard deviation for each calculated point (red diamonds). These data were extracted by the average of every c'/c ratio for all the conformations studied in this research, taking into consideration the percentage of β -amyryn in these structures.

Minor revisions

q1 - The authors must briefly introduce the structure of terpenoids and triterpenoids in the Introduction section for the reader's clarity.

A brief structural description of terpenes and triterpenoids has been added to the introduction section.

q2 - In the Methodology section (2.3. Raman spectroscopy), the authors have described the Raman excitation wavelength used in this study as 785 nm, which is generated using Nd:YAG laser. But the fundamental of Nd:YAG laser and its second harmonics emit at ~1064 and 532 nm wavelengths, respectively. Therefore, the authors must clarify whether they have used a 785 nm diode laser or second harmonics of Nd:YAG laser (532 nm) as Raman excitation wavelength.

This is a recurrent mistake in Raman spectroscopy considering the spread use of 1064 nm line and, hence, the Nd:YAG laser. The text has been corrected indicating that the laser used is a diode laser for 785 nm line.

q3 - The authors mention (Results and Discussion section, Page no. 2) that non-electronic interactions drive the assembly of homo-/hetero-oligomers of amyryns. However, it's not clear whether the author means non-electrostatic or hydrophobic interactions. It must be clarified.

The text has been corrected and updated to clarify that intermolecular attraction forces are non-electrostatic interactions.

Appendix B

Reviewer 2

The authors have answered all the reviewers' comments and incorporated all the necessary modifications in the revised manuscript. Therefore, I recommend this manuscript for publication as it is.

Thank you for the feedback.

Reviewer 3

q1 - The reviewers mention in their reply to the reviewers that that clarified "non-electronic interactions" to mean "non-electrostatic," but it remains "non-electronic" in the manuscript.

The manuscript has been properly corrected.

q2 - It appears from the equation for RBE that it is computed from the potential energies of the dimers and the trimers compared to the monomers. Those energies are reported in Table 1 in kcal/mol. I therefore found it confusing when in page 4 there is suddenly a discussion of free energy changes reported in kJ/mol. Are those really computed free energies (e.g., based on frequency calculations)? Are RBEs in table 1 therefore all free energies or potential energies? This is somewhat unclear.

The methodology section has been updated with an explanation of the calculation of ΔG for the selected trimers. These data were obtained from frequency calculations and it relate the free energy of the trimer with the dimer-monomer set.

q3 - For binding energy calculations, it is usually important to account for basis set superposition error. However, in these cases, it could be possible to assume that BSSE would be similar for the different interactions and since the focus is on relative binding energies.

As the reviewer comments, because of the relative character of this parameter, the Basis Set Superposition Error is not strictly necessary. Indeed, this parameter was used in previous research by our group with feasible results.

q4 - I recommend including the xyz coordinates of key (e.g., lowest energy) optimized structures in the supporting information for the sake of reproducibility, along with their absolute energies.

This information has been added in Dryad Digital Repository.